# High Sorption and Selective Extraction of Actinides from Aqueous Solutions

**DOI:** 10.3390/molecules26237101

**Published:** 2021-11-24

**Authors:** Linfa Bao, Yawen Cai, Zhixin Liu, Bingfeng Li, Qi Bian, Baowei Hu, Xiangke Wang

**Affiliations:** 1School of Life Science, Shaoxing University, Huancheng West Road 508, Shaoxing 312000, China; baolinfa@usx.edu.cn (L.B.); caiyawen1993@163.com (Y.C.); hbw@usx.edu.cn (B.H.); xkwang@ncepu.edu.cn (X.W.); 2Power China Sichuan Electric Power Engineering Co., Ltd., Chengdu 610041, China; libing7826450@163.com; 3Shaoxing ZeYuan Science Technology Ltd., Shaoxing 312000, China; bianqi@zykj-ztech.com

**Keywords:** actinides, species, sorption, reduction, microstructures

## Abstract

The selective elimination of long-lived radioactive actinides from complicated solutions is crucial for pollution management of the environment. Knowledge about the species, structures and interaction mechanism of actinides at solid–water interfaces is helpful to understand and to evaluate physicochemical behavior in the natural environment. In this review, we summarize recent works about the sorption and interaction mechanism of actinides (using U, Np, Pu, Cm and Am as representative actinides) on natural clay minerals and man-made nanomaterials. The species and microstructures of actinides on solid particles were investigated by advanced spectroscopy techniques and computational theoretical calculations. The reduction and solidification of actinides on solid particles is the most effective way to immobilize actinides in the natural environment. The contents of this review may be helpful in evaluating the migration of actinides in near-field nuclear waste repositories and the mobilization properties of radionuclides in the environment.

## 1. Introduction

Radionuclides, especially radioactive actinides, are the major radioactive nuclides in nuclear waste, which can be a serious threat to human health if they are released into the natural environment and accumulate in living organisms. Because of the low concentration of actinides as compared with other kinds of metal ions, the efficient elimination of actinides, especially the highly selective removal of actinides, from complicated solutions is still a serious challenge.

Over the past decades, different kinds of methods such as sorption, precipitation, coagulation, ultra-filtration, photocatalytic reduction, electrochemical reduction, and biological immobilization/reduction techniques have been extensively applied for the elimination/immobilization of actinides [1,2,3]. Different methods have different advantages or disadvantages. For example, precipitation or biological immobilization are cost-effective, however, it is difficult to reduce the concentrations of radionuclides to below the legal limits. Photocatalytic reduction generates some by-products, which may be more toxic. Sorption is an effective method to eliminate radionuclides from wastewater because of its ease of operation, high sorption capacity and large scale application. Liu et al. [4] synthesized dialdehyde wastepaper (DAWP) as a cross-linking agent to immobilize persimmon tannin (PT) for U(VI) removal and found that DAWP-PT could efficiently eliminate U(VI) from wastewater. A novel biochar (SFeS@Biochar) was synthesized and applied for the elimination of U(VI) in aqueous solutions. The results showed that it could overcome some shortcomings of nanoscale ferrous sulfide and biochar, and that it also was an improved environmentally friendly material for U(VI) removal [5]. The application of biochar-based materials for the efficient elimination of actinides have been reviewed [6,7]. The results showed that biochar-based materials had high sorption capacity and that some high valence actinides could be reduced to low valence actinides through a photocatalytic reduction process, and thereby form precipitates on biochar surfaces. Materials with high sorption capacity are crucial for the application of actinides removal.

The species and microstructures of actinides at solid particles are important in understanding the physicochemical behavior of actinides at solid-water interfaces. The species of actinides (Np(IV), U(V), Cm(III)) by X-ray absorption fluorescence spectroscopy (XAFS) analysis were reviewed by Denecke [8]. The coordination chemistry property information such as the coordination number, the bond distances of actinides with other ions, the valence change of actinides (reduction or oxidation), possible precipitation, and the formation of actinide oxide nanoparticles may be achieved through XAFS spectroscopy analysis. In the last decades, advanced spectroscopy techniques have been applied extensively to measure the species of actinides on solid–water interfaces [9,10,11,12,13]. Such information is very helpful in understanding the interaction mechanism of actinides at solid–water interface as the microstructures and species of actinides on mineral surfaces affect the mobilization and bioavailability of actinides.

Besides the aforementioned results of actinide sorption on minerals, the sorption of actinides on manmade nanomaterials should be considered, as nanomaterials have a much higher sorption capacity than natural clay minerals. More importantly, the surface graft of special functional groups on nanomaterials is relatively easy, and thereby may enhance the selective extraction of actinides from complicated solutions. Wang et al. [14] firstly applied multiwalled carbon nanotubes (MWCNTs) to remove ^243^Am(III) from aqueous solutions and found that the MWCNTs had a much higher sorption capacity than natural clay minerals. Surface-adsorbed actinides were able to mobilize from the surface to the inner channel of MWCNTs with an increase in aging time and become difficult desorb from them. The application of nanomaterials in the elimination of radionuclides has been studied extensively [15]. Metal-organic frameworks have large specific surface areas and high porosities. Their high surface area and abundant functional sites may increase their sorption ability and selectivity toward actinides through post-synthetic modification or selectivity of the building blocks. The use of MOFs for the removal of radionuclides and metal ions is reviewed by Jin et al. [16] (Figure 1), Gao et al. [17] and Zhang et al. [18]. Functional groups could bind actinides and the porous channels could transfer actinides into MOFs, and thereby could fix the actinides in MOFs. The actinides could be bound by MOFs through cation ion exchange, hydrogen bond network, chemical complexation etc., which could be measured through batch experiments, advanced spectroscopy techniques such as XAFS, TRLFS, Raman and computational calculation. The functionalization of stable MOFs is a new method for the application of MOFs in actinide sequestration. Covalent organic frameworks (COFs) are chemically stable and have excellent structural regularity. The orderly porous structure provides abundant porous channels and high surface area, which make them suitable for the efficient elimination of actinides from aqueous solutions. The application of COFs for the high selective removal of actinides is reviewed by Liu et al. [19]. The 3D MnO_2_@COF was applied for the removal of U(VI) from aqueous solutions, and the results showed ultra-fast U(VI) extraction by forming strong complexes [20]. The MOFs, COFs and their based composites showed high sorption capacity and selectivity for the efficient elimination of actinides from complicated solutions.

In this review, we summarize recent works about the surface interaction of actinides (using Am, Cm, Np, Np, Pu, U as representative actinides) on natural clay minerals and man-made nanomaterials. Advanced spectroscopy techniques such as time-resolved lase fluorescence spectroscopy (TRLFS) and X-ray absorption fluorescence spectroscopy (XAFS) are applied to understand the interaction of actinides at solid–water interfaces. Theoretical calculation was also applied to simulate the interaction of actinides on solid surfaces, which is an important method to evaluate the interaction mechanism.

## 2. Interaction Mechanism of Actinides at Solid-Water Interfaces

### 2.1. Uranium

As one of the most important radionuclides, uranium is inevitably released into the natural environment during mining process and treatment of spent nuclear fuel. The sorption of uranium from water is not only important environmental, but also for nuclear fuel extraction. Zhao et al. [21] synthesized UiO-66 and functionalized the UiO-66 with carboxyl functional groups to achieve its mono-carboxyl (UiO-66-COOH) and di-carboxyl (UiO-66-2COOH) derivatives. The -COOH groups can efficiently reduce the surface charge and thereby increase the sorption capacity of UiO-66 at low pH values. At pH 3, the sorption of U(VI) on UiO-66-2COOH achieved >100 mg/g whereas no sorption occurred on UiO-66. Dynamic column sorption experiments showed that 99% U(VI)/Eu(III) could be efficiently removed from solutions, and then recollected by dilute HNO_3_ solution. Carboxylated UiO-66 can remove U(VI) efficiently from acidic solutions (pH 3). UiO-66-2COOH may be a suitable material for the separation of actinides from wastewater. Yang et al. [22] prepared Fe-Nx-doped MOFs and applied them to the removal of U(VI) from wastewater. The adsorption-electrocatalysis results showed that FeNx centers can reduce U(VI) to U(V), before the unstable U(V) is re-oxidized to U(VI) in the presence of Na^+^ to form solid Na_2_O(UO_3_·H_2_O)_x_, which can efficiently extract U(VI) from seawater with a sorption capacity of 1.2 mg·g^−1^ within 24 h. The Na_2_O(UO_3_·H_2_O)_x_ can be removed by the electrode easily for collection, and can be converted to UO_2_ by chemical/thermal treatment, which is important for the preparation of nuclear fuel (Figure 2).

Photocatalytic reduction of radionuclides is considered one way for the in-situ solidification and immobilization of radionuclides [23]. Photocatalytic reduction of U(VI) to U(IV) is an efficient method to extract U(VI) from aqueous solutions, especially at low concentrations. Zhang et al. [24] synthesized SnO_2_/CdCO_3_/CdS (SCC) and applied it to the photocatalytic reduction of U(VI) under visible light conditions. The U(VI) could be directly reduced to U(IV) by the photo-generated electrons of SCC rather than the free radicals. The CdS offered the photoelectrons and holes, the CdCO_3_ enhanced the separation of photoelectrons and holes, whereas SnO_2_ stabilized CdS against the photocorrosion. The SCC could effectively extract U(VI) from complicated wastewater, which is a promising material for the selective preconcentration of U(VI) at low concentrations. Qiu et al. [25] fabricated a ZIF-8/g-C_3_N_4_ photocatalyst for the photoreduction of U(VI) from aqueous solutions, and found that ZIF-8/g-C_3_N_4_ exhibited high photocatalytic activity, fast photocatalytic rate and superior chemical stability for the reduction of U(VI) to U(IV). Li et al. [26] reported an eco-friendly method for the selective and rapid extraction of U(VI) under visible light conditions without using a catalyst and in which the U(VI) could be reduced to U(IV) quickly. The selective extraction of U(VI) in the presence of different kinds of metal ions showed the excellent selectivity of U(VI) extraction. The theoretical calculation indicated that the ligand-to-metal charge transfer with U(VI) contributed to U(VI) reduction to U(IV) through disproportionation reaction. The photocatalytic reduction of U(VI) to U(IV) under visible light conditions without any kinds of catalyst is helpful to understand the in-situ solidification of U(VI) in the natural environment.

The sorption of U(VI) is an efficient method to remove U(VI) from aqueous solutions, especially at relatively high concentrations. The in-situ photocatalytic reduction of U(VI) to U(IV) is also a good technique for the solidification of U(VI) from complicated solutions, especially at extra-low concentrations. The high selective elimination of U(VI) through selective adsorption-electrocatalysis processes maybe a suitable method for the extraction of U(VI) from wastewater, especially at high salt conditions.

### 2.2. Curium

Curium (Cm) is a very highly toxic radionuclide, which is created by the successive neutron capture of Am and Pu isotopes. Cm is also an important contributor to the long-term radiotoxicity of spent fuel. Knowledge about the interaction of Cm isotopes, such as ^244^Cm and ^242^Cm, at solid–water interfaces is helpful to evaluate the physicochemical behavior of Cm in the environment. The interaction of Cm(III) with kaolinite showed the formation of Cm–silicate complexes and the ternary kaolinite-Cm-silicate colloidal silicate species formed on kaolinite surfaces. The Cm–silicon complexation is independent of the kaolinite materials. The silicon influences the ligand-field of Cm(III) on mineral surfaces and in aqueous solutions. A ternary complex with dissolved silicates is formed on kaolinite surface when silicon concentration in solution is high enough [27]. The sorption of Cm(III) to silica in the presence and absence of humic acid (HA) showed that HA obviously affected Cm(III) sorption. The addition sequences of Cm(III) and HA to silica suspension at different pH values on Cm(III) sorption to silica were quite different. The sorption of Cm(III) was enhanced at low pH values (pH < 5) but decreased drastically in the mediate pH values (pH 6.5~8) in the presence of HA. The formation of silica–Cm–HA, Cm–HA–silica ternary complexes on silica surface, or Cm–HA soluble complexes in solution-affected Cm(III) sorption on silica, is dependent on the interaction of HA with silica, interaction of soluble HA with Cm(III) in solution, or the interaction of Cm(III) with surface adsorbed HA on silica [9]. In the natural environment, the presence of humic substances (HS) should be considered to evaluate the environmental behavior of Cm(III) or other kinds of actinides as HS can form strong complexes with actinides. The competition among the soluble HS in solution, surface adsorbed HS and solid particles with actinides affects the binding of actinides to solid particles. The formation of HS–actinide colloids also affects the behavior in aqueous solutions and transport in the environment.

The species of Cm is relatively useful to understand the interaction mechanism, which can be achieved from spectroscopy analysis. The high fluorescence spectroscopy property of Cm(III), which is a representative element of trivalent actinides, has been studied by TRLFS spectroscopy analysis. The TRLFS spectra is pH-dependent, and the formation of the inner-sphere surface complexation of Cm-D aspoensis is characterized by emission spectra and fluorescence lifetime. The lifetime of 68 μs at pH 3.0 corresponded to 9 water molecules in the first coordination sphere, whereas the lifetime of 162 μs suggested 3 water molecules in Cm(III) first hydration shell [28]. The biosorption interaction of Cm(III) with the functional groups of the cell showed no incorporation of Cm(III) into the cell, but with the organic phosphate groups of the cell in the outer membrane. The Cm(III) ions mainly formed strong complexes or precipitation on the cell wall. The biosorption of Cm(III) on *Rhodotorula mucilaginosa BII-R8*, which was isolated from bentonite deposits (Almeria, Spain), was studied by TRLFS and the chemical toxicity of Eu(III), as the inactive analogue of Cm(III), was observed by scanning transmission electron microscopy-high angle annular dark field (STEM-HAADF) (Figure 3) [29]. The pH-dependent TRLFS spectra of Cm(III) in the *R. mucilaginosa BII-R8* cell (Figure 3C) showed two coordination species of Cm(III) with the functional groups of *R. mucilaginosa BII-R8* cell membranes. One species is characterized at emission wavelength of 599.6 nm with the luminescence lifetime of 240 μs, whereas the other species is characterized at emission wavelength of 601.5 nm with the luminescence lifetime of 123 μs. The two species are assigned to the binding-with-phosphoryl sites and carboxyl sites of the *R. mucilaginosa BII-R8* cell membranes. The cellular localization of Cm(III) accumulation by the *R. mucilaginosa BII-R8* cell was tested using Eu(III) as the inactive analogue because Eu(III) and Cm(III) had very similar chemical properties. The STEM-HAADF image showed the electron-dense accumulation of Eu(III) at the cell surface (Figure 3B). The element distribution mapping suggested that Eu(III) mainly interacted with phosphorus, which was in good agreement with the TRLFS analysis. The results are useful to understand the interaction of actinides with bacteria, especially the binding of actinides to bacteria. It is well known that bacteria is ubiquitously present in the natural environment, so the interaction of actinide with bacteria should be considered to evaluate the behavior of actinides as bacteria also interacts with solid particles.

### 2.3. Neptunium

Neptunium (Np) is one of the most important actinides in nuclear waste. Its high toxicity and long half-life of ^237^Np (2.13 × 10^6^ year) make it a major contributor of the total radiation in spent nuclear fuel. ^237^Np could be present in radioactive wastes over an extended time. The sorption of NpO_2_^+^ at water–mineral surface is of particular importance to evaluate the high toxic and long-lived actinides in the geosphere. The interaction of Np(V) on crystalline gibbsite and amorphous Al(OH)_3_ was investigated by batch technique and advanced spectroscopy techniques. The mononuclear inner-sphere surface complexes were formed irrespective of the atmospheric conditions. The XAFS analysis suggested the formation of bidentate surface complexes (NpO_2_CO_3_^−^) with a Np-C bond distance of 2.92 Å on Al(OH)_3_ surfaces. The Al-hydroxides could effectively retard the dissemination of Np(V) ions in groundwater system (Figure 4A). The interaction of Np(V) with gibbsite or Al(OH)_3_ should be considered to evaluate the migration of Np(V) in the natural environment [30].

The reduction and sorption of ^237^Np(V) on titanium-substituted magnetite (Fe_3−x_Ti_x_O_4_) nanoparticles under reducing conditions were investigated by XAFS technique [31]. The batch sorption results indicated that the increase of Ti contents increased the sorption and reduction of Np(V) at low pH values. The HRTEM image of Fe_3−x_Ti_x_O_4_ provided no precipitation of NpO_2_ nanoparticles on the solid surfaces, which was in good agreement with the XAFS results. The Ti substitution caused sufficient number of tetrahedral iron sites to increase Np(V) sorption. The bond distances of Np–Fe and Np–Ti were ~3.45 Å and 3.64 Å respectively. The edge regions of the Np sorption Fe_3−x_Ti_x_O_4_ samples are shown in Figure 4B. The lack of Np–Np scattering peak suggested the reduction of Np(V) without precipitation after Np(V) was adsorbed to the Fe_3−x_Ti_x_O_4_ nanoparticles. The reaction of Np(V) on hematite–water interfaces were also investigated by XAFS and time resolved IR spectroscopy techniques. The results showed the formation of single monomer Np(V) inner-sphere surface complexes at the bidentate edge-share sites. The Np–Fe bond distance of 3.73–3.74 Å was found for binary edge-sharing complexes [32]. For the safe disposal of nuclear waste, the sorption of Np(V) on hematite should be considered. Np(V) could strongly contribute to the radiation inventory because of the long half-life of Np-237. The iron oxides are of particular importance in natural environmental scenarios. The strong Np(V) inner-sphere bidentate edge-sharing complexation suggests the high retardation ability of hematite for Np(V) migration in the environment. The retention of Np(V) by siderite (an Fe(II) carbonate mineral, which is relevant to near-field high-level nuclear waste repositories) was studied under anoxic conditions and the Np(V) sorption samples were measured by XAFS (Figure 5) [33]. Np(V) formed NpO_2_-like nanoparticles under anoxic conditions through the reduction of Np(V) by biotite and chlorite with reduced structural Fe(II), which is in good agreement with the formation of UO_2_-like nanoparticles for U(VI) sorption on mackinawite [34] and magnetite [35]. The strong interaction of Np(IV) with carbonate did not prevent NpO_2_-like nanoparticle precipitation, which was attributed to the reduction of Pu(V) to Pu(IV) by Fe(II) hydroxocarbonate chukanovite. The single electron required for Np(V) reduction was provided by the dissolved Fe(II) ions. The formation of NpO_2_-like nanoparticles in the presence of siderite should be considered to evaluate the physicochemical properties of near-field geological repositories.

The sorption and reduction of Np(V) on illite surface was studied under Ar atmosphere and the sorption data were simulated by two-site protolysis non-electrostatic surface complexation and cation exchange (2 SPNE SC/CE) model [36]. Np(V) sorption increased with the decrease of redox potential, and the interaction of Np(V) with illite was attributed to Np(V) reduction to Np(IV) at illite surface. The 2 SPNE SC/CE model was a reliable simulation model to evaluate the safety of actinides in deep-field geological nuclear waste repositories. The reduction of U(VI) and Np(V) in groundwater in the presence of metallic iron was analyzed by resonant inelastic X-ray scattering spectroscopy, and the results indicated the reduction of U(VI) to U(IV), and Np(V) reduction to Np(IV) on iron surface [37]. The reduction of U(VI) and Pu(V) in the environment by metallic iron unambiguously suggested the oxidized U(VI) and Pu(V) in the nuclear waste repository would be reduced to immobile U(IV) and Pu(IV) on the corroding iron in the near-field nuclear waste repository, which retarded the migration of the mobile actinides in the packages.

The oxidized actinides are generally more mobile in the natural environmental than the reduced species. Therefore the reduction of Np(V) to insoluble Np(IV) by bacteria is considered a useful method for the immobilization of Np(V). The bacterial reduction of Np(V) has been reported, but the reduction mechanism is still unclear [38]. The reduction of Np(V) alone by *Shewanella putrefaciens* could not remove Np(V) from solution. However, the addition of phosphate-producing bacterium could result in the precipitation of Np(IV). The biological reduction of Np(V) following the precipitation of Np(IV) has been observed by sulfate-reducing bacteria [39]. Through the addition of citrate, the reduction of Np(V) can be accelerated and the reduced Np(IV) can form a polycitrate complex, something which was attributed to the effect of the organic chelators on bio-reduction of Np(V). The citrate enhanced the reduction rate of Np(V) by *S. oneidensis* cell [40]. To evaluate the physicochemical behavior of Np(V) in the natural environment, the effect of humic substances, bacteria, pH, ionic strength and minerals should be considered.

### 2.4. Americium

Americium imposes a major risk for geological repositories because of its radioactivity and long-half life. Under reducing conditions, actinides such as Am and Cm are presented as trivalent cations. The ionic radii of Am and Cm elements with 6-fold coordination oxygen (r-Cm(III) = 0.97 Å and r-Am(III) = 0.98 Å) are larger than other typical cations in clay octahedral sheets such as r-Mg(II) = 0.72 Å, r-Fe(II) = 0.78 Å [41]. From the crystal chemistry, the actinides can substitute the cations such as Mg^2+^ or Fe^2+^ ions at the clay octahedral sheets. The structural substation at the octahedral sites should be understood from batch results and spectroscopy analysis. The coprecipitation/sorption of Am(III) on hectorite was studied by XAFS technique. The XAFS data for the samples of Am(III), AmCopHec, and AmAdsBru and polarized XAFS (P-XAFS) for AmCopBru at different angles (α) of 10°, 35° and 80° were measured. The P-XAFS spectra displayed very weak angular dependence (Figure 6). The Fourier transforms (FTs) were weakly affected by the change of α, i.e., the amplitude of FT contribution only decreased slightly with the increase of α, whereas the contribution at R + ∆R at 3–4 Å did not show the dependence of angular value. For AmAdsHec, the Am in the first coordination sphere with the Am–O bond distance of 2.42 Å and coordination number of 7.2 with oxygen are shorter and lower than those of Am(III) in aqueous solutions. The coordination number of 5.8 in AmCopHec is lower than that of AmAdsHec, suggesting that Am was not predominantly surface sorbed on AmCopHec but occluded in the bulk through the coprecipitation on hectorite. Am(III) ions can form monomeric inner-sphere complexes at the platelet edges at structurally distinct sites [42].

The EXAFS and Fourier transformed (FT) spectra of Am(III) on ferrihydrite at pH 8.0 and 5.5 is shown in Figure 7. The two samples exhibited the FT peak at R value of 3.2 Å, and the bond distance of Am–O was calculated to be 2.48 ± 0.02 Å for pH 5.65 and 8.0, suggesting a difficult to distinguish Am–O interaction at ferrihydrite surface. The coordination number of Am(III) with oxygen is about 6.2 ± 0.1 for both pH values. The transformation products of ferrihydrite at pH 8.0 and 5.5 were also characterized by XAFS spectroscopy. The Am–O bond distance did not obviously change, and the XAFS spectrum of the transformed sample did not change as compared with the XAFS spectrum of the Am(III) sorption sample, suggesting the stable structure of Am(III) on ferrihydrite [43]. The coprecipitation/sorption of Am(III) on magnetite was studied as a function of pH and contact time. The XAFS analysis showed the coordination of Am(III) with 7 oxygen atoms at the Am–O bond distance of 2.44 Å, and the incorporation of Am(III) at Fe structural sites at the surface of magnetite was found to form linear Am-O-Fe bonds. After two years of aging time, the bond distance and coordination number of Am(III) with oxygen in the first shell clearly decreased, suggesting the change of the structural reorganization and crystallochemical environment of Am(III) on magnetite surface. The Fe(II) shell in the aged sample was also located at slightly shorter bond distance, and participated in the cooperation with Am(III) at the octahedral brucite-like sites [44].

### 2.5. Plutonium

Plutonium is an important component of spent fuel and poses environmental health risks because of its high radioactivity and long half-life (t_1/2_ = 24,110 year). The environmental behavior of Pu is dependent on its species and valence. Pu is generally present as Pu(IV) and Pu(V) in the natural environment. Pu(V) shows high mobilization ability, whereas Pu(IV) is more immobile because of its high affinity with minerals. The reduction of Pu(V) to Pu(IV) is a good method to reduce the mobility of Pu(V). The reduction of Pu(IV/V) to Pu(IV), and Np(VI) to Np(V) by humic substances has been found by Perminova et al. [45]. The silica gel coated with HA was tested for Np(V) and Pu(V) sequestration under anaerobic conditions. The sorption of Pu(V) reached up to 97% and that of Np(V) reached up to 60%, whereas the sorption of Pu(V) and Np(V) to pure silica was lower than 20%. The humic substances significantly improved the uptake of actinides on metal oxides and thereby increased the sequestration efficiency for Pu(V) and Np(V). The reduction of Pu(V) to Pu(IV) by natural humic substances showed the reduction in the order of humic acid < fulvic acid < low molecular weight fraction. The formation of soluble complexes of Pu(IV) with low molecule weight and mobile fulvic acid affected the mobility of Pu(IV) in the natural environment. The formation of the low solubility humic substances with Fe, Mn and other kinds of hydroxides binds favorably to Pu on the hydroxides to immobile and accumulate Pu with humic substances in soils [46]. Paul et al. [47] synthesized bi-functional polymer and applied it to a preconcentration of Pu(VI) from solutions, and the results showed that the bi-functional polymer had better ability than the polymer containing either a sulfonic group or a phosphate group. The preconcentration of Pu(IV) by the bi-functional polymer was highly reproducible for the assay of Pu(IV) in the presence of other trivalent actinides. The sorption and surface-mediated reduction of Pu(V/VI) to form crystalline nanoparticles of PuO_2 + x_ · *n*H_2_O on hematite with the particle sizes of ~1.5 nm was investigated by XAFS and HRTEM methods. The formation of the nanoparticles influenced the transport of Pu crystalline nanoparticles in the environment because of the strong interaction of the Pu nanoparticles on hematite surfaces [48]. The diglycolamide-functionalized polypropyleneimine diaminobutane dendrimers were prepared and applied to the selective extraction Am(III) and Pu(IV) from high level radioactive solutions [49]. The resin showed high selective sorption of Pu(IV) and Am(III) over ((VI), Sr(II) and Cs(I) in a very wide range of acidity, i.e., 0.01~6 mol/L HNO_3_. More importantly, the adsorbed Pu(IV) and Am(III) was desorbed efficiently using buffered solution of 0.05 mol/L EDTA and 1 mol/L guanidine carbonate. The resin was used for selective separation of Pu(IV) and Cm(III) from other kinds of radionuclides in a wide acidity range. Graser et al. [50] applied capillary electrophoresis for the separation of Pu(III), Pu(IV), Pu(V) and Pu(VI) species using inductively coupled plasma sector field mass spectrometry (CE-ICP-SF-MS) at extra low concentrations (limits of 10^−12^ mol/L) (Figure 8). The CE-ICP-SF-MS is a promising separation method for the separation and detection of different Pu redox species.

## 3. DFT Calculation

The experimental investigation of actinides sorption at mineral-water interface should be carried out in glove box because of their high toxicity and radioactivity such as Am-241,243, Np-237, Cm-245,246 etc. More importantly, it is not allowed to carry out actinide experiments. Thereby, theoretical simulation of actinides behavior on mineral surface is crucial and helpful to understand the properties of actinides in the environment.

The sorption of U, Np and Pu on germanene and silicene surfaces was studied by density functional theory (DFT) simulation to calculate the energetic, geometrical and electronic properties. The optimized solvated structures of the actinides with OH^−^, NO_3_^−^ and CO_3_^2−^ on silicone surfaces were shown in Figure 9. The DFT calculation indicated that OH^−^ and CO_3_^2−^ ligands in the silicene sheet are the most strongly sites for the sorption of the actinides, whereas the NO_3_^−^ ligand has the highest affinity at the hollow binding sites of hexagonal ring of silicene. The H site is the most favorable for the sorption of actinides on germanene cluster. The strong interaction of actinides with silicene is due to the formation of Si-O-Actinide complexes [51].

The sorption of Eu(III), Pu(III), Am(III) and Cm(III) on hematite (001) surface was compared by DFT calculation. The trivalent actinides were adsorbed on the hydroxylated hematite (001) surface in vacuum to form tridentate-binuclear complexes. The ferromagnetic substrate was more semiconduction for the trivalent actinide sorption. In the sorption process, the oxygen as the bridging atom could result in the deprotonation reaction, and H^+^ was removed from the hematite (001) surface to form the hematite-*O*-An(III) configuration complex. The lowest sorption energy showed a three-fold binding geometry and formed tridentate-binuclear complexes with 6 coordination numbers. The sorption energies, geometries and electronic properties of the trivalent Eu(III) are very similar (Figure 10) [52], suggesting Eu(III) is an appropriate analogy for the trivalent actinides in the natural environment. The structures, energetics and electronic properties of U(VI) adsorbed on TiO_2_ surfaces were calculated by DFT investigation. After introduce of carbonato and hydroxo ligands, the sorption energy became more positive, indicating the weaker U(VI) sorption onto TiO_2_ nanoparticles [53]. The sorption of U(VI) on phenanthrolineamide group grafted 3D cubic mesoporous (KIT-6DAPhen) was studied by experiments and theoretical calculation. The KIT-6DAPhen showed excellent selectivity and high sorption capacity of U(VI) (328 mg/g at pH 5), and the DFT calculation indicated the reaction of KIT-6DAPhen + [UO_2_(H_2_O)_5_]^2+^ + NO_3_^−^ = [UO_2_(KIT-6DAPhen)(NO_3_)]^+^ + 5H_2_O [54]. The sorption of Np(V), Pu(V) and Pu(VI) on C_60_O surface by DFT calculation showed that the outer-sphere surface complexes are comparable with the inner-sphere surface complexes because of the hydrogen bonding. The △G and △E values of Pu(VI) complexes on C_60_O surface are more negative than those of Np(V) and Pu(V), suggesting that Pu(VI) is more easily and efficiently adsorbed on C_60_O surface than Np(V) and Pu(V) [55]. The computational simulation is an essential method to understand the interaction mechanism of actinides at solid-water interface. The interaction energy, the structures, structure-property relationship, the coordination number, and the electronic property can be calculated. The information about the interaction of actinides with different functional groups is helpful to understand the selectivity of the functional groups for the binding of actinides. The reaction coordinate energies could provide valuable information for the possible reactions and species in the sorption dynamic processes.

## 4. Competitive Sorption

Considering the presence of numerous actinides in wastewater, the competitive sorption of actinides from aqueous solution is helpful to evaluate the selective removal of target actinides. Zhou et al. [56] studied the competitive sorption of U(VI) and Th(IV) on triphosphate-crosslinked magnetic chitosan resins from solution and found that the sorption capacities of U(VI) and Th(IV) clearly decreased as compared with a single sorption system, suggesting that the U(VI) and Th(IV) competed on the same sorption sites of the resins. Neumann et al. [10] used surface X-ray diffraction and in-situ AFM to study Th(IV) sorption on mica and the results showed a linear decrease of Th(IV) sorption with increase of electrolyte cations, indicating the competitive effect among Th(IV) and electrolyte cations on mica. Virtanen et al. [57] investigated the sorption of Eu(III), Y(III) and Cm(III) from solutions, and the results showed that sorption of Eu(III) and Cm(III) sorption was obviously decreased and sorption pH-edge shifted to higher pH when the concentration of competing Y(III) was higher than 10^−4^ mol/L. The Y(III) ions occupied most surface sites, and thereby the competition among the trivalent metals decreased the sorption of actinides. Luca et al. [58] used polyacrylonitrile-phenolic resin for the adsorption of Gd(III), Am(III), Np(III), U(VI), Pu(V) from acidic solutions, and found that Pu(V) could be selectively removed from solution in the presence of lanthanides and U(VI), and minor actinides (i.e., Am(III), Np(III)), which could selectively extract Pu(V) from complicated solutions. Falaise et al. [59] applied aluminum-based MOFs for the sorption of Th^4+^, [UO_2_]^2+^ and Nd^3+^ from solutions and found that Th^4+^ was quickly adsorbed by MOFs with high selectivity, suggesting Th^4+^ could be extracted in the presence of other kinds of actinides and lanthanides. According to our knowledge, the competitive sorption of actinides from wastewater is still scarce. The high selective sorption of target actinides from complicated solutions is still difficult, something which mainly depends on the properties of actinides, the solution conditions, the surface functional groups and the structures of materials. Besides the sorption technique, the in-situ precipitation through sorption-reduction, crystallization or filtration, the selective removal of actinides from strong acidic solutions is more important as the actinides are generally presented in strong acidic solutions during the treatment process of spent fuel.

## 5. Conclusions and Perspective

In this review article, we summarized the recent works about the interaction of actinides on clay minerals, oxides and nanomaterials. The species and microstructures at molecular level were evaluated from advanced spectroscopy analysis and computational calculations. The reduction of actinides from high valence to low valence through (photo) catalytic reduction was considered an efficient method for the immobilization of actinides on solid particles, especially at extra low concentrations.

The high radioactive toxicity and long half-life of actinides result in risks to environmental and human health. In the whole processes of nuclear energy utilization, the release of actinides into the natural environmental is inevitable. The selective extraction of actinides from complicated wastewater is still one main challenge. Special functional groups of grafted materials that have high stability, high sorption ability, and excellent selectivity for target actinides so they can be extracted at low concentrations should be developed.

## Figures and Tables

**Figure 1 molecules-26-07101-f001:**
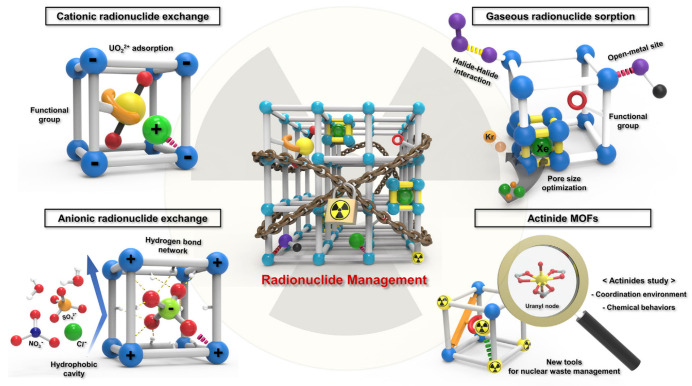
Schematic representation of applications of MOFs in radionuclide administration. The different color balls represent different atoms, the yellow ball is U, green ball is Xe and red ball is O. The other balls and sticks are mainly metal-organic frameworks (MOFs). Reprinted with permission from ref. [16]. Copyright 2021 Elsevier.

**Figure 2 molecules-26-07101-f002:**
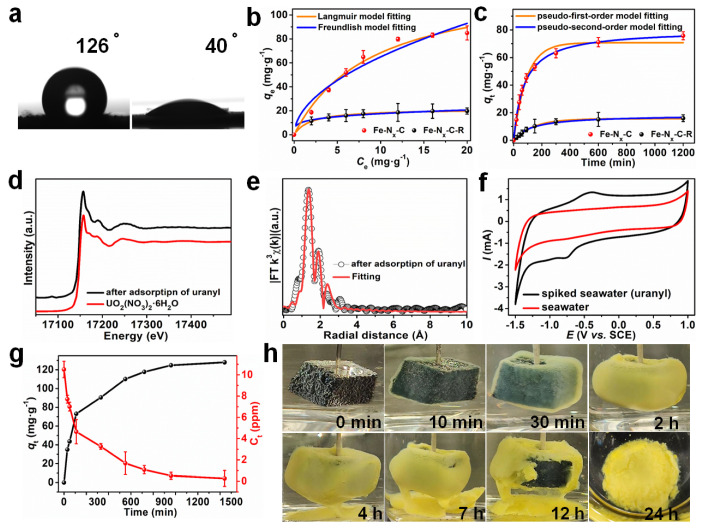
(**a**) Contact angles for deionized water on pressed pellets of nitrogen-doped carbon capsules supporting iron single-atom sites (Fe–N_x_–C) (left) and polypropylene amidoxime coated Fe–N_x_–C (Fe–N_x_–C–R) (right). (**b**) Equilibrium adsorption isotherms for uranyl ion adsorption on different materials at a fixed material–to–solution ratio of 0.1 mg·mL^−1^ in uranyl–spiked seawater (from 0 to 20 ppm). (**c**) Uranyl ion adsorption kinetics on different materials at an initial UO_2_^2+^ concentration of 10 ppm in uranyl–spiked seawater. (**d**) U L_III_–edge XANES spectra for Fe–N_x_–C–R after adsorption of uranyl, and UO_2_(NO_3_)_2_·6H_2_O. (**e**) U L_III_–edge EXAFS R-space and corresponding fitting curves for Fe–N_x_–C–R after adsorption of uranyl. (**f**) Cyclic voltammograms for uranyl-spiked seawater and natural seawater. (**g**) Uranium extraction from spiked seawater with initial uranium concentrations of ~10 ppm, using Fe–N_x_–C–R as an adsorbent electrocatalyst. (**h**) Photographs of the Fe–N_x_–C–R electrode in spiked seawater (initial uranium concentration of ~1000 ppm) during electrocatalytic extraction Reprinted with permission from ref. [22]. Copyright 2021 John Wiley and Sons.

**Figure 3 molecules-26-07101-f003:**
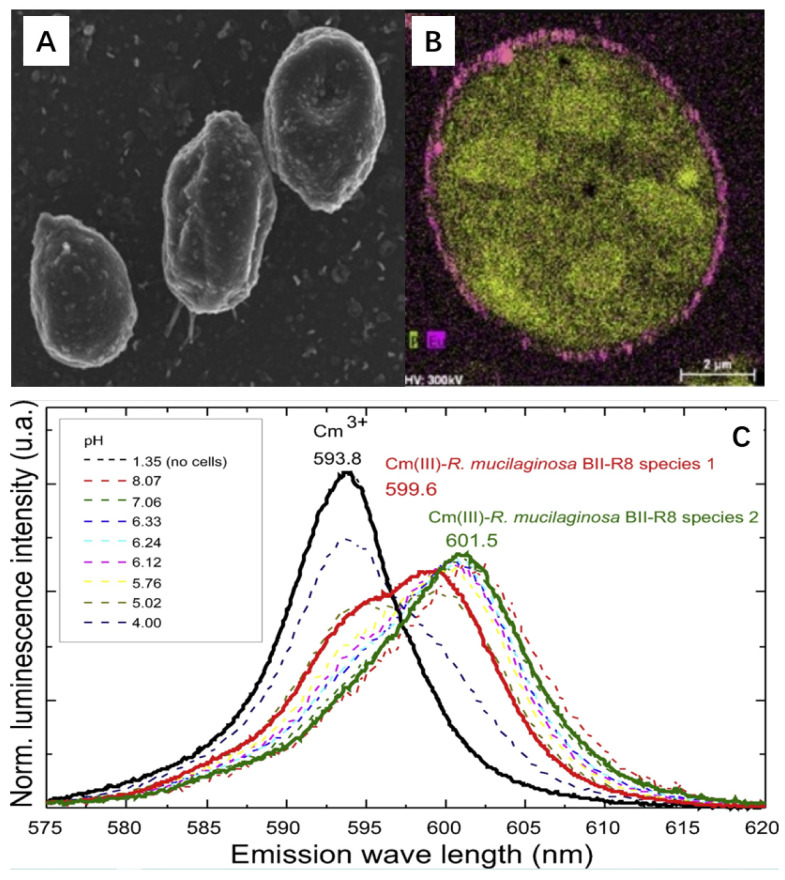
SEM image of *R. mucilaginosa* BII-R8 (**A**), phosphorus (light green) and Eu(III) (pink color) mapping image in STEM-HAADF micrographs of thin sections of *R. mucilaginosa* BII-R8 treated with 1 mM Eu (III) for 48 h (**B**), and luminescence emission spectra of 0.3 μM Cm(III) in 0.1 M NaClO_4_ measured as a function of pH at a fixed biomass concentration of 0.45 g_dry weight_/L (**C**). Reorganized from TOC. Reprinted with permission from ref. [29]. Copyright 2021 Elsevier.

**Figure 4 molecules-26-07101-f004:**
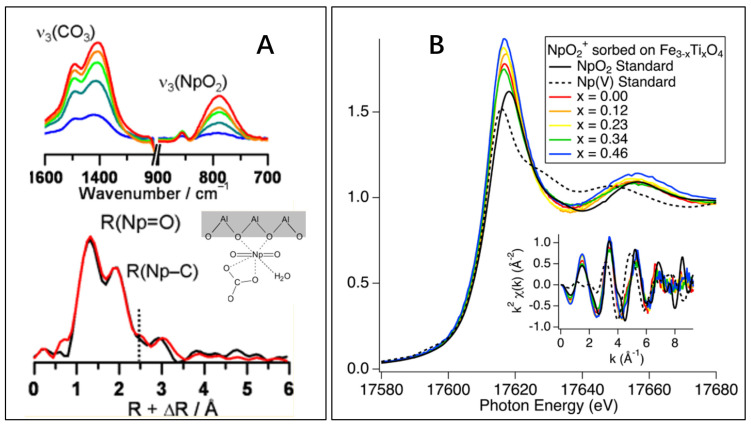
(**A**): The Np IR and EXAFS spectra and the microstructure of Np bidentate surface complexes on Al(OH)_3_ surfaces. Reorganized from TOC. Reprinted with permission from ref. [30]. Copyright 2021 Amrican Chemical Society; (**B**): normalized X-ray absorption spectra from samples, NpO_2_ and Np(V)(aq) standards and k-space EXAFS (inset). The similarity of the titanomagnetite samples’ absorption edges to the Np(IV) standard, note the lack of “neptunyl” shoulder, and lack of Np-Np scattering peak at 3.7 Å, this shows a reduction of Np(V) without precipitation Reprinted with permission from ref. [31]. Copyright 2021 Amrican Chemical Society.

**Figure 5 molecules-26-07101-f005:**
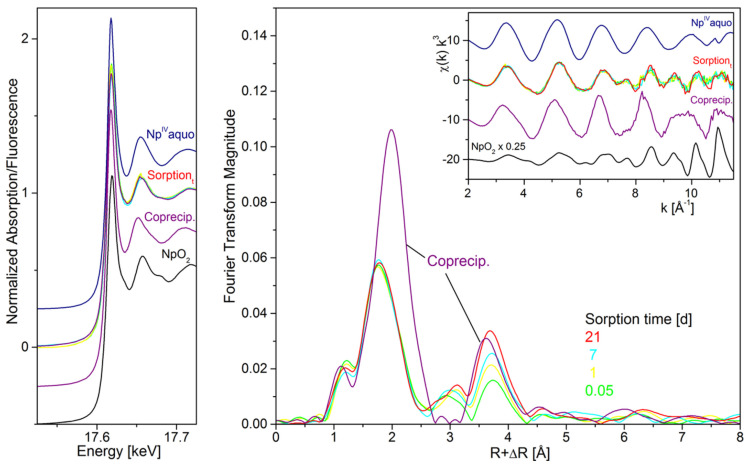
Neptunium L_3_–edge XAFS spectra of selected siderite sorption and coprecipitation samples (pH7.7 ± 0.3) along with Np references. Left: XANES spectra, right: EXAFS Fourier transform magnitude and corresponding χ(k) spectra as insert. Reprinted with permission from ref. [33]. Copyright 2021 Amrican Chemical Society.

**Figure 6 molecules-26-07101-f006:**
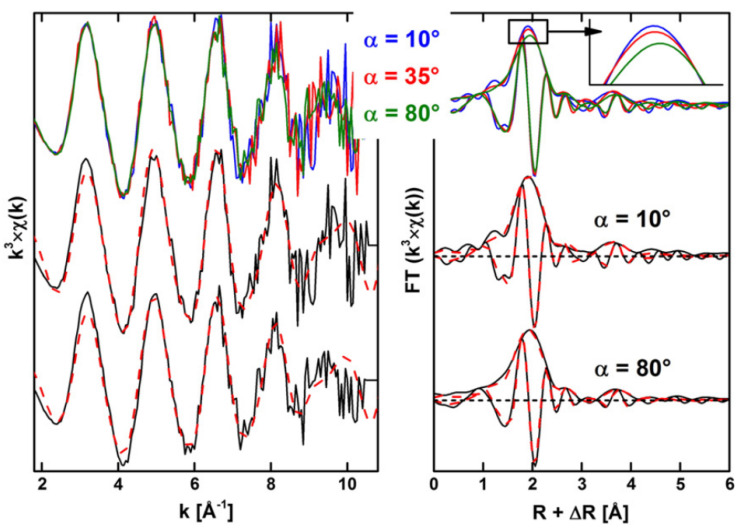
Polarized EXAFS spectra (**left**) and the corresponding Fourier transforms (**right**) of AmCopBru. Reprinted with permission from ref. [42]. Copyright 2021 Elsevier.

**Figure 7 molecules-26-07101-f007:**
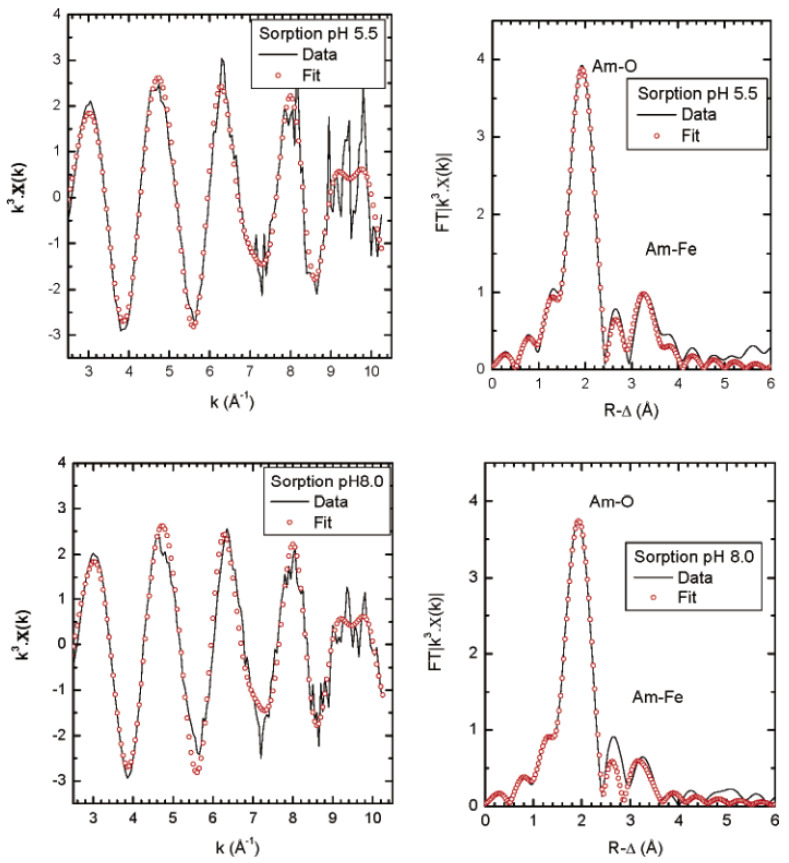
Am sorption onto ferrihydrite at pH 5.5 and 8.0. (**Left**): k^3^—weighted Am L3 edge experimental EXAFS and back transformed R–space fit results. (**Right**): corresponding Fourier transformed spectra. Experimental data are lines and fit data are circles. Reprinted with permission from ref. [43]. Copyright 2021 Amrican Chemical Society.

**Figure 8 molecules-26-07101-f008:**
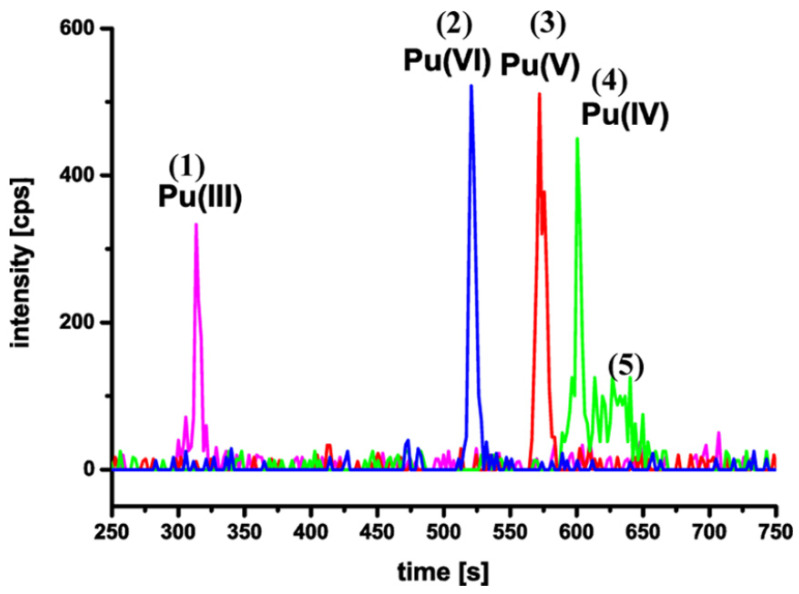
Combined electropherogram of ^242^Pu reference solutions in 1M HClO_4_: peak (1) Pu(III), peak (2) Pu(VI), peak (3) Pu(V), peak (4) Pu(IV), and peak (5) Pu(IV) polyspecies (*t* > 600 s); ([^242^Pu] = 2.1 × 10^−10^ M; hydrodynamic injection: 10 s/2 psi; voltage: 30 kV; BGE: 1 M HAc; pH = 2.4). Reprinted with permission from ref. [50]. Copyright 2021 Amrican Chemical Society.

**Figure 9 molecules-26-07101-f009:**
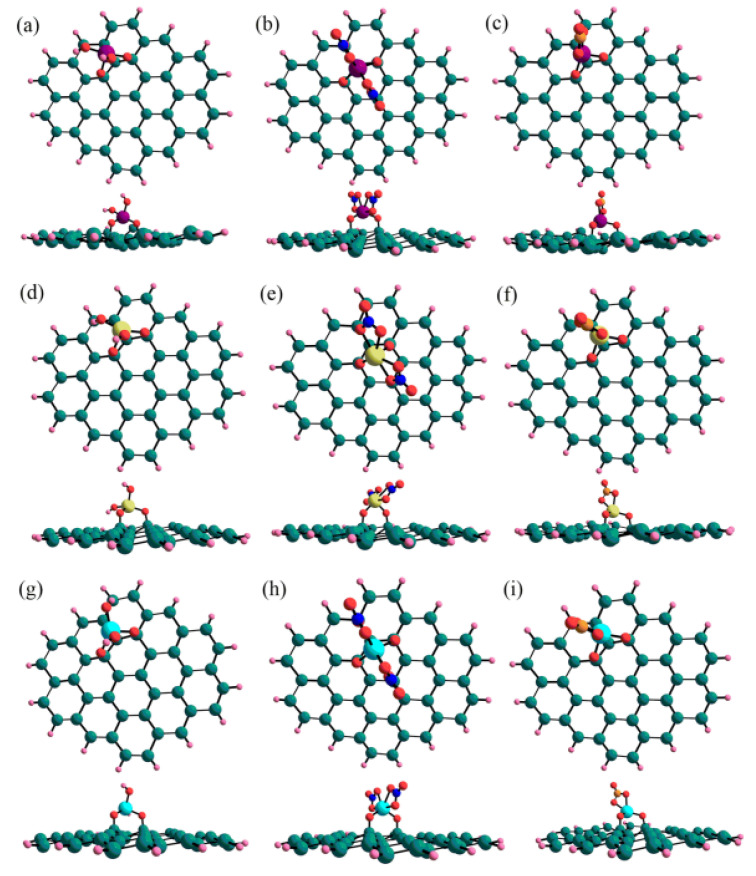
Top and side views of the optimized solvated structures in water for (**a**) [UO_2_(OH)_2_], (**b**) [UO_2_(NO_3_)_2_], (**c**) [UO_2_(CO_3_)], (**d**) [NpO_2_(OH)_2_], (**e**) [NpO_2_(NO_3_)_2_], (**f**) [NpO_2_(CO_3_)], (**g**) [PuO_2_(OH)_2_], (**h**) [PuO_2_(NO_3_)_2_], and (**i**) [PuO_2_(CO_3_)] on the Si_42_H_16_ flake. Purple means U, light green is Np, light blue is Pu, red is O, dark blue is N, orange is C, turquoise is Si and pink is H. Reprinted with permission from ref. [51]. Copyright 2021 Amrican Chemical Society.

**Figure 10 molecules-26-07101-f010:**
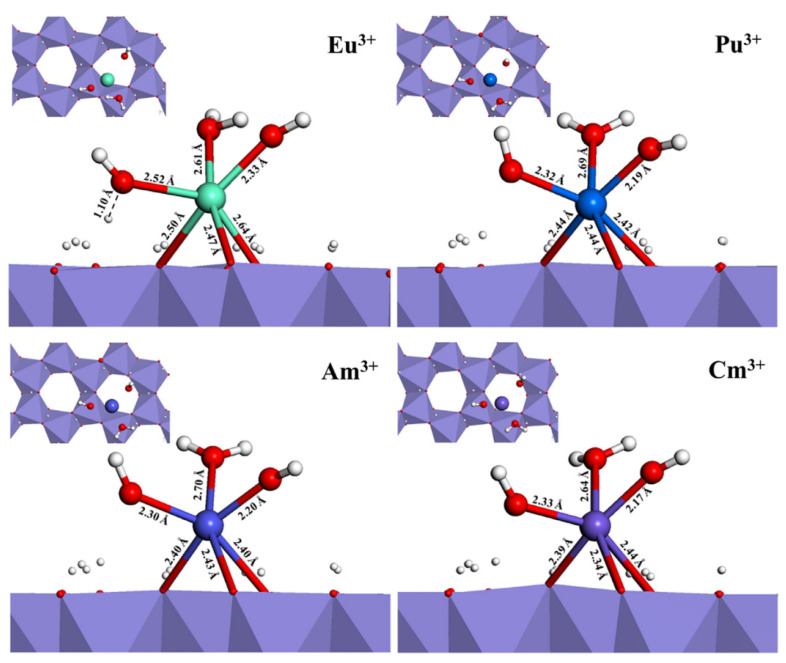
Ground state configuration of the [Eu/An(OH)_2_H_2_O]^+^ complexes adsorbed on the hematite (001) surface, viewed along the [110] direction. Adsorption geometry with the lowest sorption energy for each cation is shown. Light blue is Eu, dark blue is Pu, blue purple is Am, purple is Cm, red is O, and milky white is H in order. Reprinted with permission from ref. [52]. Copyright 2021 Elsevier.

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
