# Peer review of "High Sorption and Selective Extraction of Actinides from Aqueous Solutions"

_molecules, 2021, doi:10.3390/molecules26237101_

Round 1
Reviewer 1 Report
- We do not know the meaning of different color balls and sticks in Figure 1. Please indicate which color is which atom. In addition, the characteristic adsorption mechanisms of the MOFs should be mentioned in the introduction using the knowledge of Figure 1 as a supporting information.
- What is Fe-Nx-C, Fe-Nx-C-R, and ZIF-8/g-C3N4? Some additional information will be required in the manuscript for readers.
- Pink color means Eu(III) in Figure 2. However, we do not know the meaning of light green part.
- The yellow line and number in Figure 5 are invisible.
- Is ZIF-8/g-C3N4 a nanomaterials? If so, it is no problem.
- In the purpose of this study, focusing on the surface interaction between actinides and natural clay minerals or man-made nanomaterials is mentioned. Is Figure 3 reported about R. mucilaginosa important information for this topic?
- In Figure 9, we do not know which color is which atom.
- [Eu/Au(OH)2H2O]+ should be [Eu/Au(OH)2H2O]+ in Figure 10. Similarly, we do not know which color is which atom.
- The abbreviation of the journal title " Coordination chemistry reviews " is " Coord. Chem. Rev. ". Please modified with references, No. 8, 16.
Author Response
- We do not know the meaning of different color balls and sticks in Figure 1. Please indicate which color is which atom. In addition, the characteristic adsorption mechanisms of the MOFs should be mentioned in the introduction using the knowledge of Figure 1 as a supporting information.
Reply: The different color balls mean different atoms, such as the yellow ball is U, green ball is Xe and red ball is O. The other balls and sticks mainly are metal−organic frameworks (MOFs). The meaning of different color balls and sticks in Figure 1 have been added in Figure caption. The characteristic adsorption mechanism of MOFs have been added in revised form. “The functional groups could bind actinides and the porous channels could transfer actinides into MOFs, and thereby could fix the actinides in MOFs. The actinides could be bound by MOFs through cation ion exchange, hydrogen bond network, chemical complexation etc., which could be measured through batch experiments, advanced spectroscopy techniques such as XAFS, TRLFS, Raman and computational calculation.”
- What is Fe-Nx-C, Fe-Nx-C-R, and ZIF-8/g-C3N4? Some additional information will be required in the manuscript for readers.
Reply: The “nitrogen−doped carbon capsules supporting iron single−atom sites (Fe−Nx−C) (left) and polypropylene amidoxime coated Fe−Nx−C (Fe−Nx−C−R)” have been revised and added in Figure caption of Figure 2. ZIF-8 is a kind of metal−organic framework. C3N4 is carbon nitride.
- Pink color means Eu(III) in Figure 2. However, we do not know the meaning of light green part.
Reply: I think it should be Figure 3. Pink color is Eu(III) and light green part is phosphorus. We have revised it in the revised form with the Figure caption as “R. mucilaginosa BII-R8 (A), phosphorus (light green) and Eu(III) (pink color) mapping image in STEM-HAADF micrographs (B) and Luminescence emission spectra of 0.3 μM Cm(III) in 0.1 M NaClO4 measured as a function of pH at a fixed biomass concentration of 0.45 gdry weight/l (C)”.
- The yellow line and number in Figure 5 are invisible.
Reply: We have download Figure 5 with high quality from the paper and added in the revised form. We cannot change the color of the lines.
- Is ZIF-8/g-C3N4 a nanomaterials? If so, it is no problem.
Reply: Yes. It is a kind of metal−organic framework / carbon nitride nanomaterial.
- In the purpose of this study, focusing on the surface interaction between actinides and natural clay minerals or man-made nanomaterials is mentioned. Is Figure 3 reported about R. mucilaginosa important information for this topic?
Reply: R. mucilaginosa is a kind of bacteria. The presence of R. mucilaginosa is very important for the interaction of actinides with materials. Thereby, we added some results about the effect of bacteria on the interaction of actinides with materials.
- In Figure 9, we do not know which color is which atom.
Reply: We have given the explanation in the Figure caption with “Purple means U, light green is Np, light blue is Pu, red is O, dark blue is N, orange is C, turquoise is Si and pink is H”.
- [Eu/Au(OH)2H2O]+ should be [Eu/Au(OH)2H2O]+ in Figure 10. Similarly, we do not know which color is which atom.
Reply: Thank you very much. It is really correct that [Eu/Au(OH)2H2O]+ should be [Eu/Au(OH)2H2O]+. We have revised it in the revised form. About the color for different atom, we have given the explanation in the Figure caption with “Light blue is Eu, dark blue is Pu, blue purple is Am, purple is Cm, red is O, and milky white is H in order”
- The abbreviation of the journal title " Coordination chemistry reviews " is " Coord. Chem. Rev. ". Please modified with references, No. 8, 16.
Reply: Thanks. We have revised them and checked all the References.
Reviewer 2 Report
This review article summarizes progress made in the elimination and immobilization of actinides using a variety of techniques ranging from sorption to photocatalytic rduction. The work goes in depth to discuss specific actinides' interaction mechanisms with various solid surfaces and does an adequate job describing batch level experiments and spectroscopic/microscopic evidence to support the claims.
The review article is descriptive and useful for those looking for specific actinides to remove/treat. However, natural systems contain a complex range of these elements in mixture. This article does not present a coherent set of arguments as to how numerous actinides may be treated in solutions where competitive sorption is at play. This hinders the work from being truly excellent in its applicability as a review paper. In addition, there is no discussion on implementation of these scientific approaches to treat and remove actinides from environmental solutions beyond the bench-top. As a result, this review paper falls into a category of "useful yet not significant".
Addressing competitive sorption processes or implementation of these processes in a scalable technology should improve the impact of this review paper and its ability to truly assess the progress made in sorption technology. Indeed, sorption in this article is discussed as an applicable technology to improve society and environmental health. It only implies such impact but needs to directly address methods/apparatuses that can in fact go beyond batch+spectroscopic characterization.
Author Response
Reply: Thank you very much for your comments. This is a review article about the sorption and interaction mechanism of representative actinides (using U, Np, Pu, Cm and Am as representative actinides) on natural clay minerals and man-made nanomaterials. The competitive sorption of numerous actinides is not reviewed as few research work in the competitive sorption of actinides. About the removal of actinides from aqueous solutions, the sorption capacities of different materials are quite different, and the selective sorption is dependent on the surface functional groups. We mainly discussed the interaction mechanism in this review. The removal of actinides from solutions was not discussed in this review. In the revised form, we added some discussion about the competitive sorption of actinides, and the methods/apparatuses were also added in the revised form as “Considering the presence of numerous actinides in wastewater, the competitive sorption of actinides from aqueous solution is helpful to evaluate the selective removal of target actinides. Zhou et al. [57] studied the competitive sorption of U(VI) and Th(IV) on triphosphate-crosslinked magnetic chitosan resins from solution and found that the sorption capacities of U(VI) and Th(IV) decreased obviously as compared single sorption system, suggesting the U(VI) and Th(IV) competed on the same sorption sites of the resins. Neumann et al. [58] used surface X-ray diffraction and in-situ AFM to study Th(IV) sorption on mica and the results showed a linear decrease of Th(IV) sorption with increase of electrolyte cations, indicating the competitive effect among Th(IV) and electrolyte cations on mica. Virtanen et al. [59] investigated the sorption of Eu(III), Y(III) and Cm(III) from solutions, and the results showed that sorption of Eu(III) and Cm(III) sorption was obviously decreased and sorption pH-edge shifted to higher pH when the concentration of competing Y(III) was higher than 10-4 mol/L. The Y(III) ions could occupy most surface sites, and thereby the competition among the trivalent metals decreased the sorption of actinides. Luca et al. [60] used polyacrylonitrile-phenolic resin for the adsorption of Gd(III), Am(III), Np(III), U(VI), Pu(V) from acidic solutions, and found that Pu(V) could be selectively removed from solution in the presence of lanthanides and U(VI), and minor actinides (i.e., Am(III), Np(III)), which could selectively extract Pu(V) from complicated solutions. Falaise et al. [61] applied aluminium-based MOFs for the sorption of Th4+, [UO2]2+ and Nd3+ from solutions, and found that Th4+ could quickly adsorbed by MOFs with high selectivity, suggesting Th4+ could be extracted in the presence of other kinds of actinides and lanthanides. According to our knowledge, the competitive sorption of actinides from wastewater is still scarce. The high selective sorption of target actinide from complicated solutions is still difficult, which mainly depends on the properties of actinides, the solution conditions, the surface functional groups and structures of materials. Besides the sorption technique, the in-situ precipitation through sorption-reduction, crystallization or filtration. The selective removal of actinides from strong acidic solutions is more important as the actinides are generally presented in strong acidic solutions during the treatment of spent fuel process.”
Round 2
Reviewer 1 Report
I have no questions.
Reviewer 2 Report
The authors address comments on competitive sorption appropriately. Despite significant literature here, the authors attempt to bridge the gap by citing relevant studies on competitive aqueous complexes. I thus believe this manuscript does more than just "review" the current field, but instead also contributes to where the field may go in regards to selectivity research.